# Early recognition and management of maternal sepsis in Pakistan: a feasibility study of the implementation of FAST-M intervention

Sheikh Irfan Ahmed ![ORCID],[1] Ghulam Kubra Rind,[1] Raheel Sikandar,[2] Amir Raza,[1] Bakhtawar M Hanif Khowaja,[1] Fahmida Parveen,[2] Sehrish Khan,[2] Nazia Memon,[2] Arshia Jahangir,[3] Daayl Naim Mirza ![ORCID],[4] James Cheshire,[5] Catherine Louise Dunlop ![ORCID],[5] Sadia Shakoor,[6] Rubina Barolia,[7] Lumaan Sheikh,[1] Sarah Saleem ![ORCID],[8] Arri Coomarasamy,[5] David Lissauer[9,10]

For numbered affiliations see end of article.

**Correspondence to**
Sheikh Irfan Ahmed;
sheikh.irfan@aku.edu

## ABSTRACT

**Objective** Maternal sepsis is the third leading cause of maternal mortality globally. WHO and collaborators developed a care bundle called FAST-M (*F*luids, *A*ntibiotics, *S*ource identification and treatment, *T*ransfer and *M*onitoring) for early identification and management of maternal sepsis in low-resource settings. This study aimed to determine feasibility of FAST-M intervention in a low-resource setting in Pakistan. The FAST-M intervention consists of maternal sepsis screening tools, treatment bundle and implementation programme.

**Design and setting** A feasibility study with before and after design was conducted in women with suspected maternal sepsis admitted at the Liaquat University of Medical and Health Sciences hospital Hyderabad. The study outcomes were compared between baseline and intervention phases. In the baseline phase (2 months), the existing sepsis care practices were recorded, followed by a training programme for healthcare providers on the application of FAST-M tools. These tools were implemented in the intervention phase (4 months) to assess any change in clinical practices compared with the baseline phase.

**Results** During the FAST-M implementation, 439 women were included in the study. 242/439 were suspected maternal infection cases, and 138/242 were women with suspected maternal sepsis. The FAST-M bundle was implemented in women with suspected maternal sepsis. Following the FAST-M intervention, significant changes were observed. Improvements were seen in the monitoring of oxygen saturation measurements (25.5% vs 100%; difference: 74%; 95% CI: 68.4% to 80.5%; p<0.01), fetal heart rate assessment (58% vs 100%; difference: 42.0%; 95% CI: 33.7% to 50.3%; p≤0.01) and measurement of urine output (76.5% vs 100%; difference: 23.5%; 95% CI: 17.6% to 29.4%; p<0.01). Women with suspected maternal sepsis received all components of the treatment bundle within 1 hour of sepsis recognition (0% vs 70.5%; difference: 70.5%; 95% CI: 60.4% to 80.6%; p<0.01).

**Conclusion** Implementation of the FAST-M intervention was considered feasible and enhanced early identification and management of maternal sepsis at the study site.

**Trial registration number** ISRCTN17105658.

## STRENGTHS AND LIMITATIONS OF THIS STUDY

⇒ The study design provided the possibility to analyse operational, and practical factors for introducing the *F*luids, *A*ntibiotics, *S*ource identification and treatment, *T*ransfer and *M*onitoring intervention in the local setting.

⇒ The tools used in the study were adapted in the context of Pakistan before its implementation.

⇒ The study design does not account for the temporal effects, hence is exposed to possible selection and reporting bias.

⇒ This study was conducted at only one tertiary level hospital.

⇒ The intervention period was of only 4 months.

## INTRODUCTION

Maternal sepsis is 'a life-threatening condition defined as organ dysfunction resulting from infection during pregnancy, childbirth, postabortion or the postpartum period'.[1] Worldwide, it is the third most common cause of maternal mortality and represents around 11% of maternal deaths.[2] Pregnant patients and those who have recently delivered are more susceptible to maternal infections that can rapidly progress towards sepsis.[3 4]

Modelling studies, based on currently available data, observed that globally around 17 000 maternal deaths per year may be attributed to maternal sepsis and other maternal infections.[5] Approximately 5.7 million women suffered complications of maternal sepsis globally in 2017.[6] Annually, 99% of the 302 000 maternal deaths occurred in low-income and middle-income countries (LMICs),[7] with the highest rates reported in Africa and Asia.[7 8] The overall maternal mortality ratio in Pakistan is 186 per 100 000

live births in 2020,[9] and sepsis ranked as the third leading cause of maternal deaths (16.3%),[10 11] following haemorrhage and eclampsia.[2]

In 2015, the global maternal and neonatal sepsis initiative was introduced by the WHO and other partners to design strategies that could help in improving the recognition and management of maternal and neonatal sepsis in low-resource settings.[12] This initiative aimed to develop and test the effective strategies to prevent, detect and manage the maternal and neonatal deaths caused by sepsis.[12] This included the key recommendation to develop a sepsis care bundle specific to the maternal population.

The use of care bundles in the management of sepsis has been found to reduce mortality and improve patient outcomes in high-income countries.[13] Most notably, the surviving sepsis campaign bundles,[14 15] as well as the UK Sepsis Trusts' Sepsis six bundle,[16] have been widely used.[15 16] Despite this, there was no sepsis care bundle in particular for the maternal population which can be implemented efficiently in low-resource settings.[17–19]

To address this, a care bundle for treatment of maternal sepsis in low resource settings was developed after reaching the consensus utilising a modified Delphi approach by an international health expert panel.[20] The selected components were Fluids, Antibiotics, Source identification and control, Transfer to an appropriate level of care and ongoing Monitoring of mother and neonate. The bundle was named 'FAST-M' as a memorable acronym for supporting implementation and communication.[20] The FAST-M bundle was first implemented in 15 government healthcare facilities in Malawi, and it was both found to be feasible to implement and also resulted in improved clinical care.[21]

Despite existence of national sepsis guidelines for Pakistan, their uptake remains low especially for obstetric patients.[22] Additionally, there is uncertainty that the existing evidence-based approach to prevent, recognise and treat maternal sepsis is effective and implementable in a low-resource setting. It was, therefore, planned to determine the feasibility of the FAST-M intervention in the local setting before recommending its implementation at scale. The study protocol and procedures have been described in detail and published elsewhere.[23]

To optimise the use of the FAST-M intervention in Pakistan, a qualitative study was conducted to understand the existing sepsis management practices/behaviours, and to identify potential gaps and resource availability in the local setting.[24] Based on the qualitative findings described in a separate paper and availability of resources in the facility, the FAST-M bundle care tools which were used in Malawi[21] were modified to plan a practical approach for the implementation of the intervention in our local context.[24] This study aimed to determine whether the FAST-M intervention is feasible to introduce for the early detection and management of maternal sepsis in a low-resource setting in Pakistan. We hypothesise that the FAST-M intervention in this adapted form could be successfully implemented

and would then improve maternal sepsis care, including its early recognition and management.

## METHODS

A prospective feasibility study with a before and after design was conducted at the Liaquat University of Medical and Health Sciences (LUMHS) Hyderabad, Pakistan from the period of June 2021 to December 2021. Women admitted to obstetrics and gynaecology units of the facility, who were pregnant, or within 6 weeks of miscarriage, ectopic pregnancy, termination of pregnancy or delivery were considered eligible for this study. These eligible patients were monitored on Modified Early Obstetric Warning Scores (MEOWS) chart during their inpatient admission. Those who had abnormal maternal observations or healthcare practitioner's (HCP) concern regarding potential maternal infection or pregnant women with a fetal tachycardia greater than or equal to 160 beats per min were included in the study. There were no exclusion criteria for the study.

In the baseline phase, data were collected from facility record using Case Report Form-CRF 1 (facility audit form) to assess available resources for sepsis management and overall maternal outcomes in last 6 months. Data of the patients who were meeting eligibility criteria (figure 1) were recorded on CRFs 2 and 3 to assess existing practices for management of maternal sepsis. Following this, the intervention phase was carried out. At first, HCPs were trained on the use of FAST-M bundle care tools (MEOWS chart, decision tool and treatment bundle) to early identify and manage sepsis patients. The HCPs training was an ongoing process throughout the implementation of the intervention. The FAST-M bundle care tools were then implemented in the intervention phase in the facility and data of patients meeting the eligibility criteria were recorded using CRFs 2 and 3. The flow of study participants has been demonstrated in figure 1.

Comparisons were made between baseline and intervention phases to assess any change in time and frequency of vital signs monitoring (on MEOWS chart) of patients, and application of all components of FAST-M treatment bundle (online supplemental file 1) within an hour of sepsis recognition. Time 0 or start time was considered when the patient had developed one or more red trigger/s on the MEOWS chart during an inpatient stay. Same applied for the patient transferred from other facilities already on treatment. Maternal outcomes (sepsis rate and related mortality) were also compared between baseline and intervention phases.

### Intervention

The planned intervention consisted of 3 components:
1. Maternal sepsis screening tools
2. FAST-M treatment bundle
3. FAST-M implementation Programme

### Component 1: maternal sepsis screening tools
Maternal sepsis screening tools included the MEOWS and the FAST-M decision tool (online supplemental file

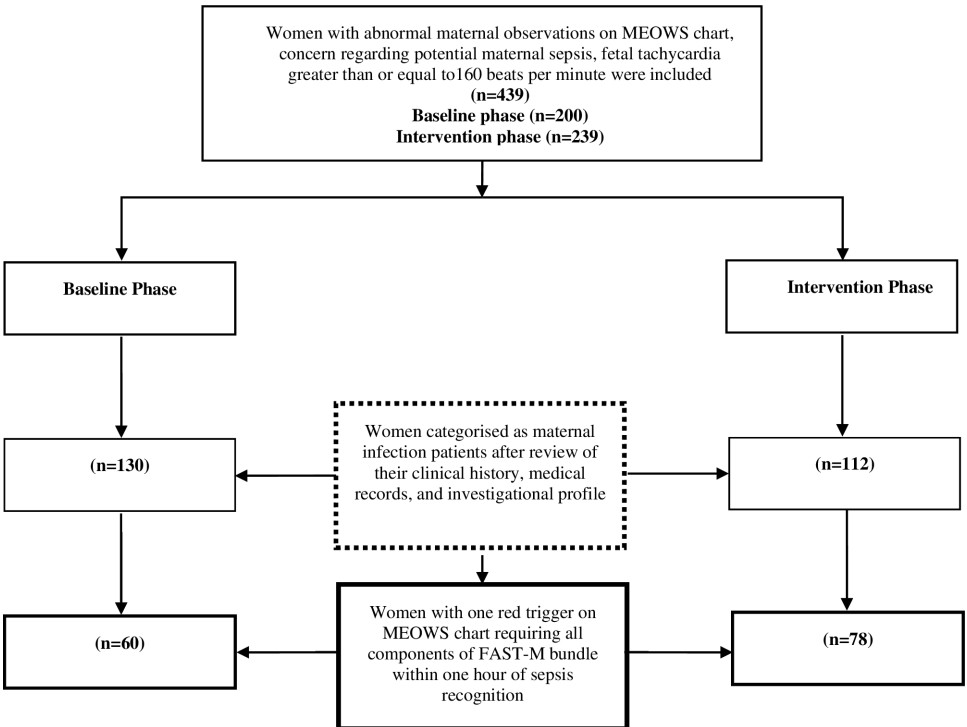

**Figure 1** Flow of participants through the study. FAST-M, *F*luids, *A*ntibiotics, *S*ource identification and treatment, *T*ransfer and *M*onitoring; MEOWS, Modified Early Obstetric Warning Scores.

1). MEOWS is the tool adapted by Malawi study based on an early warning score approach and simplified from the approach adopted in UK hospitals.[25] The tool assisted in charting an individual's physiological parameters over time with guiding thresholds that signal clinical action if they become abnormal.[26]

The MEOWS chart used in FAST-M implementation in the districts of Malawi[21] was adapted in the context of Pakistan for this study. During the adaptation phase, the HCPs recommended the inclusion of oxygen saturation into the MEOWs chart to determine patient's clinical condition considering the outbreak of COVID-19 infection and the availability of pulse oximeters at the study site. Therefore, the oxygen saturation component was incorporated in the modified MEOWS chart based on the scope of existing practices and the significance of this indicator in the identification of patient's clinical condition.

In the intervention period, eligible patients who had abnormal observations (indicated by a single red or two yellow flags) on MEOWS chart (online supplemental file 1) screened for potential sepsis using the FAST-M decision tool. Those who developed a red trigger on the MEOWS chart received an immediate clinical review. FAST-M treatment bundle was initiated within an hour on clinician's judgement on suspicion of sepsis and continued until advised to end the treatment.

Those patients who triggered two yellow flags on the MEOWS chart and were thought to have possible infection were reviewed within 3 hours. All suspected cases (those without red or two yellow triggers) were still considered for having a risk of sepsis and remained under observation for possible development of red or yellow triggers;, these patients were monitored on the MEOWS chart hourly throughout the day and were managed as per local guidelines.

### Component 2: FAST-M treatment bundle
The FAST-M treatment bundle consists of Fluids, Antibiotics, Source identification and Control, Assessment of the need to Transport/Transfer to a high level of care and Ongoing Monitoring (of the mother and neonate). The first-line and second-line antibiotic therapy guidelines were modified as per the local context during the adaptation phase[24] (online supplemental file 1). Patients with suspected sepsis were commenced immediately on the FAST-M treatment bundle within an hour of identification of sepsis. Each patient's treatment was recorded on the treatment bundle, along with any actions taken and any reasons for not completing certain elements.

### Component 3: FAST-M implementation programme
The implementation programme (figure 2) consisted of the FAST-M training, selection of clinical champions and feedback achieved from data collected during the baseline phase, and through the facility audit form.

The 2-day interactive training sessions were delivered by healthcare experts after the completion of the baseline phase and before the implementation of the FAST-M bundle intervention. Instructions on the use of FAST-M bundle care tools were provided, and hands-on training of healthcare providers was conducted during the sessions.

HCPs including doctors and nurses (n=40) working in obstetrics and gynaecology units, internal medicine,

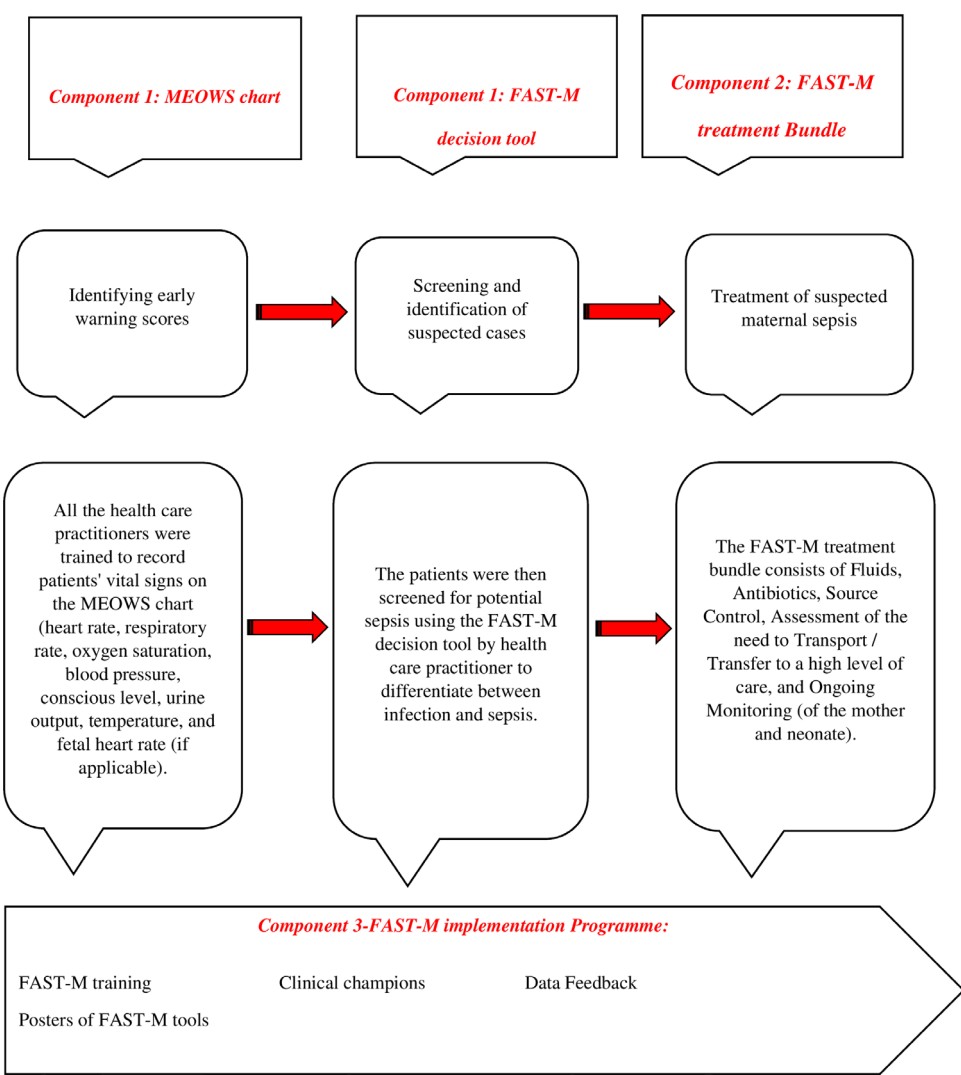

**Figure 2** Components of FAST-M intervention. FAST-M, *F*luids, *A*ntibiotics, *S*ource identification and treatment, *T*ransfer and *M*onitoring; MEOWS, Modified Early Obstetric Warning Scores.

operating rooms, emergency and intensive care units of LUMHS hospital were trained to record patients' vital signs on the MEOWS chart (heart rate, respiratory rate, oxygen saturation, blood pressure, conscious level, urine output, temperature and fetal heart rate (if applicable)). These observations were then charted on a MEOWS chart in the inpatient setting and then inserted into the patient's files. Follow-up training sessions were also conducted every 2 weeks by the local team leads during the implementation of the FAST-M bundle intervention.

The local clinical champions and team leaders were identified and trained to take the lead in implementation of the FAST-M intervention at the study site. The selected champions included doctors and nurses from different hospital departments primarily involved in the management of maternal sepsis patients including obstetrics and gynaecology units, emergency department, internal medicine, and intensive care unit.

The overarching goal of each champion was to encourage engagement and compliance with the FAST-M bundle. To achieve this goal, champions at each unit were engaged in several key activities that included disseminating knowledge, advocating, navigating boundaries, facilitating consensus, arranging meetings with stakeholders, tracking quality indicators and developing organisational communication strategies and relationships. The patient records collected were also counterchecked by the clinical champions of the assigned units. These champions also helped the study team in the identification of patients with suspected sepsis and their enrolment in the study and remained engaged throughout the implementation process.

In addition, training certificates were distributed among the participants. Moreover, FAST-M tools (MEOWS chart, decision and treatment tool) were displayed in the form of posters throughout all the units involved in the management of maternal sepsis and implementation of FAST-M intervention.

### Study period
The study comprised a baseline period of 2 months in which usual practices to detect and treat maternal sepsis

existing locally were assessed. The FAST-M training was conducted on the completion of the baseline period. Following training, an intervention period of 4 months was commenced to assess any change in practice.

## Outcomes
### Primary process outcomes
1. The proportion of patients admitted with indications of sepsis, who received appropriate monitoring (full set of vital sign measurements on admission recorded on MEOWS chart).
2. The proportion of women with suspected maternal sepsis receiving the FAST-M treatment bundle (including each bundle component) within 1 hour of identification of sepsis.

### Secondary outcomes
1. Rate of vital signs with complete recording of MEOWS chart components (heart rate, respiratory rate, oxygen saturation, blood pressure, mental state, urine output, temperature, and fetal heart rate (in pregnant women)).
2. Rate of application of FAST-M bundle components (Fluids, Antibiotics, source identification (identified clinically) and control, assessment of the need to transport/transfer to a high level of care and ongoing monitoring (of the mother and neonate)).
3. Rate of maternal morbidity and mortality after FAST-M intervention.

## Data collection
Quantitative data for this feasibility study were collected by a senior research assistant on the facility record CRF 1 (online supplemental file 2), and patient notes including the MEOWS charts, FAST-M decision and treatment tools onto the relevant paper-based CRFs 2 and 3 (online supplemental files 3,4) following patient consent (online supplemental file 5). Data were composed of various outcomes: structural, clinical, organisation and any adverse events. The data were maintained in an investigator file to be secured in a locked cabinet. Anonymised and deidentified patients' information recorded on the data collection sheet were recorded in a database located on a secure server.

During the study, biweekly monitoring of the study site was undertaken by the research coordinator, to observe practice, collect feedback and provide updates on the study site's performance, and evaluate any additional training requirements.

## Statistical analysis
Statistical Package for Social Sciences (SPSS) V.19.0 (IBM Corporation, Armonk, NY, USA) and STATA V.14 computer software (Stata Corporation, Texas, USA) were used for data processing and statistical analysis. Data were reported in frequency and percentages for categorical observations. A $\chi^2$ test for independence was used to compare components of FAST-M bundle between phases (baseline and intervention). The $\chi^2$ test for trend (p trend) was used to observe the performance during the intervention phase. Statistical significance was determined at $p \leq 0.05$.

## Patient and public involvement
Patients were not involved in the design of this study. The FAST-M intervention was implemented on the enrolled patients; however, they were not invited to comment on the study design, interpret the results or contribute to writing or editing of this document for accuracy of findings. Hospital staff participated in the implementation but were not involved in the analysis of findings.

## RESULTS
*Vital signs monitoring assessment:* 439 women were included in the study (200 in baseline phase and 239 in the intervention phase), with vital signs recorded on admission (table 1); 242/439 were cases of suspected maternal infection, and 138/242 were women with suspected maternal sepsis. Figure 1 provides the summary of the screening, and enrolment of the study participants. Following the

**Table 1** Patient's assessment of vital signs on admission (n=439)

| Vital sign | Baseline, n=200 | Intervention, n=239 | P value |
|---|---|---|---|
| Respiratory rate | 194 (97%) | 239 (100%) | 0.009 |
| Oxygen saturations | 51 (25.5%) | 239 (100%) | 0.0005 |
| Temperature | 195 (97.5%) | 239 (100%) | 0.019 |
| Heart rate | 194 (97%) | 238 (99.6%) | 0.051 |
| Systolic BP | 196 (98%) | 239 (100%) | 0.042 |
| Diastolic BP | 196 (98%) | 239 (100%) | 0.042 |
| Urine output | 153 (76.5%) | 239 (100%) | 0.0005 |
| Mental state | 196 (98%) | 238 (99.6%) | 0.182 |
| Fetal heart rate (in pregnant woman)* | 80 (58%) | 123 (100%) | 0.0005 |

*For FHR, the denominator is 138 in baseline and 123 in the intervention phase.
BP, blood pressure; FHR, fetal heart rate.

FAST-M implementation, significant improvements were seen in the assessment of oxygen saturations in the intervention phase compared with the baseline phase (difference: 74%; 95% CI: 68.4% to 80.5%; p<0.01), fetal heart rate assessment (difference: 42.02%; 95% CI: 33.7% to 50.28%; p≤0.01), measurement of urine output (difference: 23.5%; 95% CI: 17.6% to 29.4%; p<0.01). Temperature, respiratory rate, heart rate and blood pressure were measured in 97% of the cases in the baseline phase but showed an increase of almost 3% in the intervention phase (difference: 2.5%; 95% CI: 0.34% to 4.66%; p≤0.05). No significant difference was found in the assessment of mental status in the intervention phase when compared with the baseline phase (difference: 1.6%; 95% CI: −0.52% to 3.7%; p=0.18) (online supplemental file 6).

*Baseline characteristics of women with suspected maternal sepsis:* 242 women were identified by the FAST-M decision tool as fulfilling the criteria for suspected maternal infection, of which 65% (130/200) women were during the baseline phase and 46.9% (112/239) during the intervention phase. Out of 242 women with maternal infections, 138 (57%) women were identified with suspected maternal sepsis. Sixty of 138 (46.2%) women were during the baseline phase and 78 of 112 (69.6%) women were observed during the intervention phase.

The characteristics of the patients with suspected maternal sepsis and various laboratory investigations carried out for the identification of source of infection are presented in online supplemental files 7,8.

*Compliance with the FAST-M treatment bundle within 1 hour:* women with suspected maternal sepsis received all the components of the FAST-M treatment bundle, with the goal of treatment within 1 hour. The mean start time of the FAST-M treatment bundle was 30.67±15.18 min for fluids, 26.43±13.45 for antibiotics administration, 53±12.72 min for investigation of the source of infection and its treatment, 19.62±16.26 to assess the need to transfer high level and 54.43±10.1 for monitoring. Compliance improved following the intervention, as compared with the baseline (70.5%; 95% CI: 60.4% to 80.6%; p<0.01) (table 2).

Improvements in the management of sepsis cases were seen in all FAST-M bundle components, women were more likely to get IV fluids (52.2%; 95% CI: 38% to 66.3%; p<0.01), antibiotics (24.5%; 95% CI: 12.3% to 36.6%; p<0.001), source identification (27.8%; 95% CI: 15.3% to 40.3%; p<0.01), assessment of the need to transfer required to higher level of care (65.3%; 95% CI: 52.8% to 77.6%; p<0.01) and ongoing monitoring (63.3%; 95% CI: 50.3% to 76.4%; p<0.01) within an hour of suspected sepsis identification compared with the baseline phase (figure 3). The improvements in the management of women with suspected maternal sepsis were maintained throughout the intervention, with no significant decline in performance over time (p trend value>0.05) (online supplemental file 9).

*Maternal outcomes:* the facility data shows 80 (28.6%) deaths due to maternal sepsis out of total 279 total maternal deaths recorded from the period of January to December 2021 (online supplemental file 10). A 9.4% decline in the sepsis-related maternal mortality was observed: 52/159 (32.7%) in the month of January–June and 28/120 (23.3%) during the July–December 2021 (p=0.087).

## DISCUSSION
### Main findings
The implementation of the FAST-M intervention was feasible, and effective in early identification and management of maternal sepsis at the study setting. There was high compliance with all monitoring components after the intervention, with a significant increase in compliance with completion of the MEOWS charts. Following the intervention, women with suspected sepsis were more likely to receive all the critical early interventions needed to treat maternal sepsis. There was high and sustained compliance during the postinterventional observation period with completion of the FAST-M treatment bundle within an hour once maternal sepsis was suspected.

**Table 2** Completion of FAST-M bundle within 1 hour of recognition of maternal sepsis

| | Baseline (n=60) | Intervention FAST-M (n=78) | Intervention over time | | | | P value | P trend |
| | | | Month 1 (n=29) | Month 2 (n=18) | Month 3 (n=17) | Month 4 (n=14) | | |
| --- | --- | --- | --- | --- | --- | --- | --- | --- |
| Total bundle | 0 (0%) | 55 (70.5%) | 20 (69%) | 9 (50%) | 15 (88.2%) | 11 (78.6%) | 0.0005 | 0.2069 |
| Fluids | 21 (35%) | 68 (87.2%) | 24 (82.8%) | 17 (94.4%) | 15 (88.2%) | 12 (85.7%) | 0.0005 | 0.7516 |
| Antibiotics | 43 (71.7%) | 75 (96.2%) | 27 (93.1%) | 18 (100%) | 16 (94.1%) | 14 (100%) | 0.0001 | 0.3979 |
| Source identification | 41 (68.3%) | 75 (96.2%) | 28 (96.6%) | 17 (94.4%) | 17 (100%) | 13 (92.9%) | 0.0005 | 0.8405 |
| Assessment of need to transfer high level | 7 (11.7%) | 60 (76.9%) | 22 (75.9%) | 10 (55.6%) | 16 (94.1%) | 12 (85.7%) | 0.0005 | 0.1739 |
| Monitoring | 12 (20%) | 65 (83.3%) | 24 (82.8%) | 10 (55.6%) | 17 (100%) | 14 (100%) | 0.0005 | 0.0384 |

FAST-M, Fluids, Antibiotics, Source identification and treatment, Transfer and Monitoring.

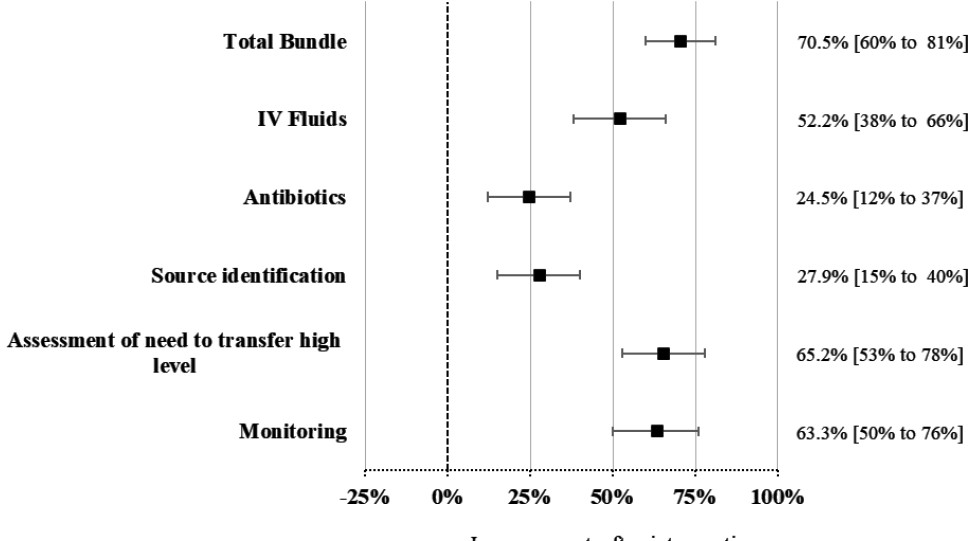

**Figure 3** Completion of FAST-M bundle within 1 hour of recognition of maternal sepsis. FAST-M, *F*luids, *A*ntibiotics, *S*ource identification and treatment, *T*ransfer and *M*onitoring.

## Interpretation

The use of the MEOWS chart has been related to improved health outcomes and has been shown to predict severe maternal morbidity and mortality, though there is limited evidence on its feasibility and impact on outcomes in low resource settings.[27] This study confirms the applicability of the MEOWS chart in a low resource setting in Pakistan. In this study, we found that with the combined use of the MEOWS chart along with healthcare provider training, resulted in improvement in the monitoring of vital signs after the implementation of the FAST-M intervention.

In low-resource settings, the lack of screening processes and resources can be an obstacle to the early detection and treatment of maternal sepsis.[28 29] It was observed that while the Royal College of Obstetricians and Gynaecologist guidelines were recommended for patient management and treatment at the LUMHS hospital, there was no standardised chart for the recording of patients' vital signs.

The MEOWS chart used in the Malawian settings[21] was adapted before its implementation in Pakistan based on healthcare providers' suggestions during the adaptation phase of the study. Consequently, the healthcare providers were trained on an adapted MEOWS chart for regular recording of a full set of vital signs to screen patients who are at risk of developing sepsis, including the recording of oxygen saturations. Providers showed a large improvement in the reliability of routine oxygen saturation recording following the intervention, so this component was shown to be a feasible addition in the Pakistan context.

Early detection of sepsis patients is critical for prompt treatment and improved maternal outcomes.[3 30 31]

Research indicates that early warning scores accompanied by a clear escalation policy guide the healthcare providers on the need for additional action to optimise patient care.[31 32] The decision tool used guided the healthcare providers on when to proceed with treating the patient as having suspected sepsis and initiating the FAST-M bundle.[33 34] As noted earlier, the FAST-M screening tools in Malawi facilitated the identification of maternal sepsis patients and guided healthcare staff to escalate management plans.[21] This study shows that it was also true in the Pakistan context where the FAST-M decision tool along with the MEOWS chart guided healthcare providers in appropriate screening and identification of women with suspected sepsis and assisted in making timely decisions for such cases.

The study was not powered to demonstrate reductions in maternal mortality or severe morbidity, but due to the critical importance of this outcome, it was prespecified as a secondary outcome. Following intervention, we observed a 9.4% decline in mortality of maternal sepsis patients, which suggests that the early identification of the sepsis patients and their timely management may as hoped lead to improved clinical outcomes. This finding is consistent with evidence from high-income countries where the use of sepsis care bundles for the immediate treatment of sepsis has been demonstrated in large observational and interventional studies to reduce mortality and improve patient's condition.[13 35] It has been reported that each part of the care bundle should be completed to reach the desired result, and failure to complete any of the components may have an impact on the overall patient outcome.[32] In the Malawi study, the FAST-M bundle implementation did not demonstrate a reduction in maternal mortality,[21] so our mortality reduction finding in Pakistan is particularly important.

Fluid administration is considered an important part of early sepsis management.[20 36] We observed a significant improvement in timely fluid administration once the FAST-M bundle was implemented. The restoration of organ perfusion is an important goal of maternal sepsis treatment and fluid resuscitation should seek to restore organ

perfusion while maintaining haemodynamic stability.[32] However, the benefits and risks of intensive early goal-guided fluid resuscitation have been the subject of intensive research.[33–36] The bundle care tools we used guided healthcare providers with a recommended fluid administration approach and to cautiously reassess the patient's haemodynamic condition between fluid boluses, through clinical examination and monitoring of the patient's vital signs. This sought to avoid overloading the patient which can actually worsen sepsis outcomes.[37 38] Higher volume of fluids in patients with sepsis and septic shock has been associated with harm, although a meta-analysis showed that the quality of evidence was very low.[38] The CLASSIC pilot study showed that restrictive fluid therapy was feasible in the ICU setting and it improved the mortality outcomes in the restrictive group, although not significantly.[39]

Fluid management in patients with maternal sepsis and pre-eclampsia is challenging due to the increased risk of pulmonary oedema and must therefore be particularly cautiously managed by healthcare professionals.[32] During the FAST-M implementation, clear teaching and guidance were provided to the healthcare providers for proper fluid administration in pre-eclamptic or eclamptic patients.

Antibiotics are considered the cornerstone of sepsis treatment, and studies have shown that timely administration of antibiotics improves patient outcomes and reduces mortality.[40 41] The risk of death from sepsis increases with each hour of antibiotic delay (OR: 1.04 per hour; 95% CI: 1.03 to 1.06; p 0.001).[40 41] As a result, empiric broad-spectrum antibiotics should be given as early as possible, and it is normally suggested that clinicians should aim to provide this where possible within 1 hour of sepsis detection.[42] In Malawi, 42.3% improvement was seen in intravenous antibiotics administration within an hour of sepsis detection, from a relatively low baseline performance of 25%.[21] It is possible that the 24.5% increase in this study in timely initiation of antibiotic administration may have led to improved clinical outcomes of women with suspected maternal sepsis.

Source identification was improved by 27.8% after FAST-M implementation in the study setting. Similar findings were reported in Malawi, where source identification improved by 18.2% within an hour of sepsis detection after FAST-M intervention.[21]

In the baseline phase, it was observed that clinicians had reservations to advise additional tests to identify the source of infection due to the logistics of undertaking these investigations in the facility and potential additional costs to be incurred by the patients. Considering this, the healthcare providers were given guidance during the training sessions regarding the significance of further investigations if the source is not initially clear with the clinical history and examination.[32] Consequently, an increase was seen in measurements of blood cultures, imaging (abdominal and chest), urine and other blood tests in the intervention phase (online supplemental file 8).

Among patients receiving treatment for maternal sepsis, the need for transfer to a higher level of care should be considered, and the decision on a patient's transfer made by a senior clinician as part of their review.[32] In Malawi, 43.9% improvement was seen in the consideration of transfer to the high level of care after the FAST-M intervention implementation.[21] During the training programme, the importance of assessment of a patient for transfer within 1 hour of recognition of sepsis was highlighted, and emphasis was made on expediting the process if their vital signs were not improving after initial sepsis treatment. Following this, the assessment of need to transfer was improved by 65.3% within an hour of sepsis detection in the intervention period.

Overall, following the FAST-M intervention, more women were monitored and screened for maternal sepsis, and the complete FAST-M treatment bundle was administered within an hour in 70.5% of women with suspected sepsis. The study findings demonstrate that the FAST-M intervention was feasible and improved the quality of care and clinical outcomes when implemented in the context of a large government hospital in Pakistan. FAST-M has the potential to be used as an integrated approach for early recognition and management of maternal sepsis in low-resource health settings of Pakistan and other LMICs. However, as we found when translating the FAST-M tools from Malawi to Pakistan, some modifications in the intervention were needed to ensure that they were optimised for the local context and available resources.

### Strengths and limitations

The strength of our research is that it is the first feasibility study conducted for the assessment of a maternal sepsis care bundle in a low-resource setting within Pakistan. This approach provided the opportunity to analyse practical, and operational factors for introducing the FAST-M intervention in the local setting. The findings of this study can lead to optimisation of the bundle prior to the large-scale intervention trial. Also, this study adds further weight to Malawi work demonstrating the feasibility of the FAST-M bundle in different low resource settings.

However, the study has several limitations. Our research design does not account for possible temporal effects, hence is exposed to possible selection and reporting bias.[43] Although a before and after design could not identify the cause-and-effect relationship, this was the practical approach for determining the feasibility of the FAST-M intervention locally.

This study was conducted at one site only that is, LUMHS which is a tertiary hospital that was comparatively better equipped than basic health units (BHUs) in the country to implement any care bundle approach. Hence, the intervention will next be implemented in other low-resource settings including BHUs across Pakistan. A large multicountry randomised trial is also underway to more robustly ascertain the effectiveness of the FAST-M bundle to improve maternal sepsis care and outcomes in LMICs.[44]

Although the intervention was found to be feasible, we recognise that the intervention period was short. A further assessment period is required to examine how well the FAST-M intervention is sustained and if it becomes normalised as a routine part of clinical care in the longer term. This additional information has been collected via an embedded qualitative study and will be reported separately.

Additionally, the high influx of patients and shortage of healthcare staff in LUMHS could have led to irregular monitoring which might have resulted in limited screening of the patients. Some of the women with suspected sepsis may have been categorised as maternal infection cases, which might have restricted the ability to demonstrate the differences between intervention and baseline phases.

Some of the referred patients in the baseline phase were observed to be already on fluid and antibiotics before admission to the hospital, which could have limited the ability of a before and after comparison, though it was only 9.4% (13/138) of patients in the baseline phase who were noted to have already received fluids and antibiotics. Additionally, the healthcare providers might have been more compliant with the FAST-M intervention, and the outcomes of this study therefore better than one would see if such an intervention was rolled out at scale due to the close monitoring of the site during the conduct of the study.

## CONCLUSION

Implementation of the FAST-M intervention was considered feasible at the LUMHS hospital. The long-term vision is that the intervention will be trialled in other settings across Pakistan, based on the outcomes of this feasibility study, and it will help to reduce the high rate of maternal deaths caused by sepsis in the country.

**Author affiliations**
[1]Obstetrics and Gynaecology, The Aga Khan University, Karachi, Pakistan
[2]OBGYN, Liaquat University of Medical and Health Sciences, Jamshoro, Pakistan
[3]Medical College, The Aga Khan University, Karachi, Sindh, Pakistan
[4]Medical College, Ziauddin University, Karachi, Sindh, Pakistan
[5]Institute of Metabolism and Systems Research, University of Birmingham, Birmingham, Birmingham, UK
[6]Pathology and Laboratory Medicine, Aga Khan University, Karachi, Sindh, Pakistan
[7]School of Nursing and Midwifery, Aga Khan University, Karachi, Sindh, Pakistan
[8]Community Health Sciences, Aga Khan University, Karachi, Sindh, Pakistan
[9]Malawi-Liverpool-Wellcome Trust Clinical Research Programme, Blantyre, Blantyre, Malawi
[10]Global Maternal and Fetal Health, University of Liverpool, Liverpool, UK

**Acknowledgements** We would like to acknowledge the healthcare providers from the LUMHS hospital Dr Prince Akash, Dr Aqsa, Dr Batool, Dr Sana, Dr Qurat ul ain, Dr Sahiba, and staff nurse Tomsina who served as Fluids, Antibiotics, Source identification and treatment, Transfer and Monitoring (FAST-M) champions and actively took on their role in implementing the FAST-M bundle intervention. We would also like to acknowledge Ms Ruqaiya Bano for data collection of this study and Ms Maliha Fazal for editing the manuscript.

**Contributors** DL and SIA designed the study and its processes. RS, FP, NM and SK provided support in data collection. AR, SIA, GKR, BMHK, AJ and DNM carried out the data analysis and interpretation. SIA, GKR, AR and BMHK took the lead in writing

the manuscript. SIA took responsibility for the overall content as the guarantor. AC, DL, CLD, JC, LS, RS, RB, SSH, SS and FP reviewed the manuscript and provided feedback on the article.

**Funding** This research was commissioned by the National Institute for Health Research (NIHR). Award Ref: NIHR300808 Host: University of Liverpool. The views expressed in this paper are those of the authors and not necessarily those of the NIHR or the Department of Health and Social Care. The project was also supported by HRP Alliance for Research Capacity Development in Sexual and Reproductive Health, University of Birmingham and Aga Khan University Research Funds.

**Competing interests** None declared.

**Patient and public involvement** Patients and/or the public were not involved in the design, or conduct, or reporting or dissemination plans of this research.

**Patient consent for publication** Consent obtained directly from patient(s).

**Ethics approval** This study involves human participants. Ethical approval for this study was obtained from the LUMHS hospital (REC/-886, 4-87), Aga Khan University Ethical Review Committee (2019-2061-7102) and National Bioethics Committee (515/20/). Participants gave informed consent to participate in the study before taking part.

**Provenance and peer review** Not commissioned; externally peer reviewed.

**Data availability statement** Data are available upon reasonable request. The data sets were collected and analysed and can be made available from the corresponding author on reasonable request.

**ORCID iDs**
Sheikh Irfan Ahmed http://orcid.org/0000-0002-8391-8559
Daayl Naim Mirza http://orcid.org/0000-0002-3396-3087
Catherine Louise Dunlop http://orcid.org/0000-0002-4792-9496
Sarah Saleem http://orcid.org/0000-0002-6797-8631

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
