## [Reviewer comments · BMJ Open]

This paper was submitted to a another journal from BMJ but declined for publication following peer review. The authors addressed the reviewers' comments and submitted the revised paper to BMJ Open. The paper was subsequently accepted for publication at BMJ Open.

ARTICLE DETAILS

TITLE (PROVISIONAL)	Early recognition and management of maternal sepsis in Pakistan: A feasibility study of the implementation of FAST-M Intervention
AUTHORS	Ahmed, Sheikh; Rind, Ghulam Kubra; Sikandar, Raheel; Raza, Amir; Khowaja, Bakhtawar; Parveen, Fahmida; Khan, Sehrish; Memon, Nazia; Jahangir, Arshia; Mirza, Daayl; Cheshire, James; Dunlop, Catherine; Shakoor, Sadia; Barolia, Rubina; Sheikh, Lumaan; Saleem, Sarah; Coomarasamy, Arri; Lissauer, David

VERSION 1 – REVIEW

REVIEWER	Dünser, Martin W. Johannes Kepler University Linz
REVIEW RETURNED	26-Oct-2022

GENERAL COMMENTS	This is an interesting study dealing with an absolutely relevant and important topic pertinent to global health care. The authors need to be congratulated on their work. As it is the nature of a review process, I have some concerns. Major: 1. My overall feeling is that the manuscript is too lengthy, particularly the Introduction and Discussion sections are much too long and must be shortened by at least 50% each. On the other hand, the Methods section could include more information (see below). The Results section must be shortened by presenting data now written out in the text in the respective tables.2. I miss important points in the Methods section: These are: How was suspected maternal sepsis defined? Please also define the time until an intervention was started or finished (e.g. starting with admission to the hospital?) Was the participating centre supplied with material (e.g. pulse oximeters) or antibiotics before the implementation period?3. More information needs to be added to Table 1 in order to better describe the patient population (e.g. demographics, comorbidities such as diabetes mellitus, arterial hypertension, pre-eclampsia or other pregnancy-related conditions, rate of C-section in patients included after delivery).4. It is unclear why Table 1 gives data on all patients with suspected infection while Table 2 gives data on patients with suspected sepsis. As I understand it from both the title and the paragraph outlining the inclusion criteria, the authors focused on
---

	patients with suspected sepsis. Therefore, I suggest that the authors only report data for this patient population. This also needs to be clearly stated in the Methods section. 5. It is furthermore unclear why maternal outcomes are reported for the entire maternal (sepsis) population in the year 2021, while the authors evaluated interventions in a smaller population. Analyse and report only one patient population! You are focusing on the feasibility of implementing the FAST-M bundle. Please also report your results accordingly. 6. The conclusion paragraph of the text is not supported by the data presented in the results section. The conclusion given in the abstract, however, is. Therefore, I suggest to use the conclusion statement as given in the abstract also at the end of the Discussion section. Minor:  1. Please include a short hypothesis statement at the end of the Introduction section. 2. In the first paragraph of the Methods section, give more details on the design of the study (retrospective data analysis, prospective analysis?) and indicate the time period during which the study was performed. 3. How did you choose the antibiotic therapy? Was the choice of the empirical antibiotic therapy based on local resistance data? This is an important point, was several studies from areas with high antibiotic resistance have shown that increasing the rate of early empirical antibiotic therapy is not affecting patient outcome if local resistance data are not taken into account. 4. Please include the time period until which interventions were started/ended quantitatively (e.g., as mean/median values with standard deviation/interquartile ranges) in Table 1.
--	---

REVIEWER	Shields, Andrea University of Connecticut, Obstetrics & Gynecology
REVIEW RETURNED	31-Dec-2022

GENERAL COMMENTS	Thank you for inviting me to review this interesting study. The authors should be commended for tackling such an important global issue. The following are my comments for the authors: Abstract Line 4 Spell out WHO when first appearing in manuscript Line 22 Recommend more detail on timeline of baseline and intervention phase in the abstract. Line 44-46 Clarify the comparison for 0 vs 70.5%. Is this the percentage of women with suspected maternal sepsis who received all components of the FAST-M treatment bundle within one hour? Strengths and limitations Line 28 and 30 Recommend replacing “which” to “this” and “that might” to “which may” Introduction Page 5 Line 10 Recommend pregnant patients Line 38 are these maternal deaths from sepsis, or overall maternal deaths?
--

Page 6
Line 53 Recommend “for” instead of “among”
Line 40-48

Page 7
Methods
Line 39-49 The authors discuss a potential limitation of the study of including patients transferred to their facility with suspicion of sepsis who are already on fluids and antibiotics. Recommend authors clarify that these patients were included in the study in this section. In the results, the authors should state how many patients in their population were in this category.
If possible, recommend authors also do a subanalysis excluding women who were transferred from another facility for suspected sepsis already receiving treatment to understand the impact this group has on the outcomes.

Line 57-58 Recommend authors define what is meant by “who are meeting eligibility criteria”

Page 7
Line 11-12 Provide more detail on which comparisons are going to be assessed for “change in monitoring and management of sepsis patients”
Line 14-15 Define which maternal outcomes are going to be assessed.

Page 9
Line 2 Recommend authors indicate where information on triggers can be found in the supplemental material.
Recommend authors define when time 0 starts (e.g., recognition of abnormal vital signs, etc.). How do they deal with those patients who are transferred from other hospitals already on sepsis treatment pathway – what is considered time 0 for these patients?
Line 55-57 Recommend more details on interactive scenarios. Were these simulations?

Page 10
Line 4-12 Which healthcare workers were eligible for training? Did all of these healthcare workers receive the FAST-M training? If not, what % of the healthcare workers completed training?

Line 41-44 Were all patients automatically enrolled in the study or did they consent to participate?

Page 11
Line 1-3 Recommend including how long the training period was here.

Line 18-21 The FAST-M includes several best practices that do not necessarily need to occur within the first hour to still be considered standard of care, e.g., transfer to a higher level of care and monitoring). The authors suggest that one of their outcomes is the proportion of women receiving each component of the bundle within one hour – did they include transfer, etc in this measurement? Please explain why these are important milestones to achieve within the 1st hour in their location/population.

Page 12

	Line 50-52 What group was the 138/242 from? Baseline? Intervention? Page 15 Line 1-9 How does the reported timeline of maternal deaths correspond to the baseline/training/implementation phase? Page 18 Line 8-18 Recommend moving this paragraph to the background/introduction.
--	---

VERSION 1 – AUTHOR RESPONSE

Reviewer: 1	Responses
Dr. Martin W. Dünser, Johannes Kepler University Linz Comments to the Author: This is an interesting study dealing with an absolutely relevant and important topic pertinent to global health care. The authors need to be congratulated on their work. As it is the nature of a review process, I have some concerns.	Thank you for your appreciation and time to review this manuscript.
1. My overall feeling is that the manuscript is too lengthy, particularly the Introduction and Discussion sections are much too long and must be shortened by at least 50% each. On the other hand, the Methods section could include more information (see below). The Results section must be shortened by presenting data now written out in the text in the respective tables.	Thank you for the suggestions. We have revised the manuscript accordingly. All changes have been tracked in the revised version submitted as a supplementary file and highlighted in the clean copy of the main manuscript.
2. I miss important points in the Methods section: These are: How was suspected maternal sepsis defined? Please also define the time until an intervention was started or finished (e.g. starting with admission to the hospital?) Was the participating centre supplied with material (e.g. pulse oximeters) or antibiotics before the implementation period?	Thank for your questions. Please see below responses:  a. Maternal sepsis was defined based on red triggers/ flags on the MEOWS chart and clinician's suspicion of maternal sepsis. Lines: 226-230 have been edited/added to explain. b. The treatment bundle started on suspicion of sepsis by clinicians and continued until their decision to end the treatment. Lines: 226 to 230 have been edited/added to explain. c. The participating centre was not supplied with any of the resources. The guidelines for intervention were adapted before its implementation based on the availability of local resources to assess its feasibility in the local setting. Line 221 has been edited to explain.

3. More information needs to be added to Table 1 in order to better describe the patient population (e.g. demographics, comorbidities such as diabetes mellitus, arterial hypertension, pre-eclampsia or other pregnancy-related conditions, rate of C-section in patients included after delivery).	Thank you for the comment, and we appreciate the rationale for requesting this addition. However, the data collection focused only on the feasibility of implementing the FAST-M intervention in the local setting, so we do not have this information available to add to the manuscript.
4. It is unclear why Table 1 gives data on all patients with suspected infection while Table 2 gives data on patients with suspected sepsis. As I understand it from both the title and the paragraph outlining the inclusion criteria, the authors focused on patients with suspected sepsis. Therefore, I suggest that the authors only report data for this patient population. This also needs to be clearly stated in the Methods section.	Thank you for the suggestion. The FAST-M intervention includes sepsis screening. This screening must be conducted on any patients with infection to ensure those with suspected sepsis are reliably identified. Therefore, to assess the feasibility of this sepsis screening component we enrolled all the eligible patients (those with any infections or possible infections) to identify those with a suspicion of sepsis. Table 1 details the vital signs monitoring of this larger group of enrolled patients (n=439) based on which 242/439 cases of suspected maternal infection, and 138/242 with suspected maternal sepsis were distinguished. This has been explained in lines 331 to 334 & Figure 1. Treatment bundle was given to 138 cases of suspected sepsis illustrated in table 2. If the editor would like this table to be moved to the supplementary materials then we can do this, but we request if possible for it to be maintained in the main manuscript as we feel making this information easily accessible to the reader will be helpful in their understanding of the work.
5. It is furthermore unclear why maternal outcomes are reported for the entire maternal (sepsis) population in the year 2021, while the authors evaluated interventions in a smaller population. Analyse and report only one patient population! You are focusing on the feasibility of implementing the FAST-M bundle. Please also report your results accordingly.	Thank you for the suggestion and apologies for the lack of clarity. The paper has been revised to make it clear, yet this study is indeed using a single study population. We conducted a facility audit in the baseline phase to assess available resources for sepsis management and maternal outcomes before the intervention period in the last six months i.e. January-June 2021, lines: 199-200 An audit after the intervention was again conducted to assess difference in outcomes with the use of FAST-M tools. The reported results are for 279 maternal deaths from sepsis in 2021 i.e., 52/159 (32.7%) in the month of January-June and 28/120 (23.3%) during the July-December 2021, lines: 376-379.
6. The conclusion paragraph of the text is not supported by the data presented in the results section. The conclusion given in the abstract, however, is. Therefore, I suggest to use the	Thank you for the suggestion. The concluding statement of the abstract has been added to the main conclusion as suggested; Lines 529-530

conclusion statement as given in the abstract also at the end of the Discussion section.	
1. Please include a short hypothesis statement at the end of the Introduction section.	Thank you. The hypothesis statement has been added as suggested; Lines 170-172
2. In the first paragraph of the Methods section, give more details on the design of the study (retrospective data analysis, prospective analysis?) and indicate the time period during which the study was performed.	Thank you. Lines 178-180 of the methods section have been revised with additional information included.
3. How did you choose the antibiotic therapy? Was the choice of the empirical antibiotic therapy based on local resistance data? This is an important point, was several studies from areas with high antibiotic resistance have shown that increasing the rate of early empirical antibiotic therapy is not affecting patient outcome if local resistance data are not taken into account.	Thank you for your query. The first and second lines of antibiotic therapy were determined in the first phase of the study (adaptation) at which point expert microbiological advice, including available local knowledge of resistance patterns was taken into account, alongside other considerations such as availability and cost were incorporated. This is described in a separate paper (reference # 24).
4. Please include the time period until which interventions were started/ended quantitatively (e.g., as mean/median values with standard deviation/interquartile ranges) in Table 1.	Thank you for the comment. The primary aim of the study was to assess feasibility of starting the FAST-M treatment bundle within one hour of sepsis suspicion, therefore we noted the date and time of the MEOWS chart recordings, the time of sepsis suspicion (development of red trigger), and the date and time of administration of fluid. The end date and time of fluid and antibiotic were not recorded as it was ended on the clinician's advice based on the patients' clinical assessment. Regarding adding more information on the median start time and interquartile range of the interventions, this has now been added in the result section: lines: 357 to 359

Reviewer: 2	Responses
Dr. Andrea Shields, University of Connecticut Comments to the Author: Thank you for inviting me to review this interesting study. The authors should be commended for tackling such an important global issue.	Thank you for your feedback and time to review this manuscript All changes have been tracked in the revised version submitted as a supplementary file and highlighted in the clean copy of the main manuscript.
Abstract	

Line 4 Spell out WHO when first appearing in manuscript	Done in Line 72. Thank you
Line 22 Recommend more detail on timeline of baseline and intervention phase in the abstract.	Done in lines 81-83. Thank you
Line 44-46 Clarify the comparison for 0 vs 70.5%. Is this the percentage of women with suspected maternal sepsis who received all components of the FAST-M treatment bundle within one hour?	Thank you for the comment. Yes, this means that in the baseline phase, none of the enrolled patients received all elements of the FAST-M bundle within one hour of recognition of maternal sepsis whereas a 70.5% improvement was seen in the intervention phase.
Strengths and limitations Line 28 and 30 Recommend replacing “which” to “this” and “that might” to “which may”	Thank for the recommendations. The statements on the strengths and limitations have been shortened and revised as suggested by the editor’s comments.
Introduction Page 5 Line 10 Recommend pregnant patients	Revised in line number 126. Thank you
Line 38 are these maternal deaths from sepsis, or overall maternal deaths?	This is the overall national rate for maternal mortality.
Page 6 Line 53 Recommend “for” instead of “among”	Revised in line 157. Thank you
Page 7 Methods Line 39-49 The authors discuss a potential limitation of the study of including patients transferred to their facility with suspicion of sepsis who are already on fluids an antibiotic. Recommend authors clarify that these patients were included in the study in this section. In the results, the authors should state how many patients in their population were in this category.	Thank you for the suggestion. On further analysis, we found out that 9.4% (13/138) of the suspected sepsis cases were referred from other facilities and were administered fluid and antibiotics before admission to the hospital. All these cases were observed only in the baseline group and not the intervention phase. Therefore, we have made revisions accordingly to clarify, lines 523-524.
If possible, recommend authors also do a subanalysis excluding women who were transferred from another facility for suspected sepsis already receiving treatment to understand the impact this group has on the outcomes.	Thank you for the suggestion. This would be helpful to understand the impact this group has on the outcomes but as mentioned above, we observed 9.4% (13/138) cases of suspected sepsis that were transferred from other facilities and these patients were found only in the baseline phase. Since there were no patients in the intervention group that were transferred from another facility, we couldn’t assess change on impact.
Line 57-58 Recommend authors define what is meant by “who are meeting eligibility criteria”	Thank you for the recommendation. Lines 180-182 define the eligibility criteria of study participants. Also, Fig.1. demonstrates the flow of patients through the study with those who meet the eligibility criteria, line 189 edited accordingly.
Page 7 Line 11-12 Provide more detail on which comparisons are going to be assessed for “change in monitoring and management of sepsis patients”	Lines 197-199: Comparisons were made between the baseline and intervention phases to assess any change in monitoring (vital signs recordings) and management of sepsis patients through the use of the FAST-M treatment bundle (supplemental file 1).

Line 14-15 Define which maternal outcomes are going to be assessed.	Included in lines 199-200
Page 9	
Line 2 Recommend authors indicate where information on triggers can be found in the supplemental material.	Thank you for the recommendation. The first tool in supplemental file 1 (MEOWS chart) demonstrates normal and abnormal readings (as red and yellow flags). Lines 226-230 been edited to explain.
Recommend authors define when time 0 starts (e.g., recognition of abnormal vital signs, etc.). How do they deal with those patients who are transferred from other hospitals already on sepsis treatment pathway – what is considered time 0 for these patients?	Thank you for the recommendation. Time 0 is when the patient had abnormal signs on the MEOWS chart (red/yellow triggers) during an inpatient stay and was enrolled in the study. Same applied for the patient transferred from other facility already on treatment. The same regime and guidelines (of the FAST-M treatment bundle) were followed for those transferred from another facility after they developed red triggers on the MEOWS chart and enrolled in the study. Yet, we found out that none of the suspected sepsis patients that were enrolled in the study were transferred from another facility already on treatment during the intervention phase.
Line 55-57 Recommend more details on interactive scenarios. Were these simulations?	No, these were interactive sessions and have been revised in the manuscript (line 252)
Page 10	
Line 4-12 Which healthcare workers were eligible for training? Did all of these healthcare workers receive the FAST-M training? If not, what % of the healthcare workers completed training?	Lines 257-258 have been revised: Health care practitioners including doctors and nurses (n= 40) working in obstetrics and gynecology units, internal medicine, operating rooms, and intensive care units of LUMHS hospital involved in the management of maternal sepsis were trained. These HCPs were selected by their head of departments and were those who deal with maternal sepsis patients. There were approximately 3 to 5 participants from each unit involved in the implementation of the intervention.
Line 41-44 Were all patients automatically enrolled in the study or did they consent to participate?	Following patient consent (highlighted in line 308)
Page 11	
Line 1-3 Recommend including how long the training period was here.	The two-day training sessions (highlighted in line 250)

Line 18-21 The FAST-M includes several best practices that do not necessarily need to occur within the first hour to still be considered standard of care, e.g., transfer to a higher level of care and monitoring). The authors suggest that one of their outcomes is the proportion of women receiving each component of the bundle within one hour – did they include transfer, etc in this measurement? Please explain why these are important milestones to achieve within the 1st hour in their location/population.	Thank you for the comment. T in the FAST-M bundle stands for assessment of the need to Transfer to a higher facility within one hour of sepsis recognition so next steps could be taken at the earliest to expedite the process if transfer is required. Lines 474-476 have been edited to explain.
Page 12 Line 50-52 What group was the 138/242 from? Baseline? Intervention?	242 patients were suspected infection patients, and 138/242 patients were categorized as suspected maternal sepsis patients based on clinician suspicion. 60/138 patients in the baseline, 78/138 in the intervention group. Table 2 shows the division of both the groups
Page 15 Line 1-9 How does the reported timeline of maternal deaths correspond to the baseline/training/implementation phase?	We divided these timelines into two phases. Jan – June 2021 (during baseline phase). July to December from the time of training till implementation of the intervention. We observed a 9.4% decline in the sepsis related maternal mortality: 52/159 (32.7%) from January-June and 28/120 (23.3%) during the July-December 2021
Page 18 Line 8-18 Recommend moving this paragraph to the background/introduction.	Thank you for the recommendation. These lines have been omitted to shorten the length of the manuscript and avoid any excessive repetitions.